# Reducing aggregation caused quenching effect through co-assembly of PAH chromophores and molecular barriers

Yinjuan Huang[1], Jie Xing[2], Qiuyu Gong[3], Li-Chuan Chen[4], Guangfeng Liu [1], Changjiang Yao[1], Zongrui Wang[1], Hao-Li Zhang[4], Zhong Chen [1] & Qichun Zhang[1]

The features of well-conjugated and planar aromatic structures make $\pi$-conjugated luminescent materials suffer from aggregation caused quenching (ACQ) effect when used in solid or aggregated states, which greatly impedes their applications in optoelectronic devices and biological applications. Herein, we reduce the ACQ effect by demonstrating a facile and low cost method to co-assemble polycyclic aromatic hydrocarbon (PAH) chromophores and octafluoronaphthalene together. Significantly, the solid photoluminescence quantum yield (PLQYs) for the as-resulted four micro/nanocrystals are enhanced by 254%, 235%, 474 and 582%, respectively. Protection from hydrophilic polymer chains (P123 ($PEO_{20}$-$PPO_{70}$-$PEO_{20}$)) endows the cocrystals with superb dispersibility in water. More importantly, profiting from the above-mentioned highly improved properties, nano-cocrystals present good biocompatibility and considerable cell imaging performance. This research provides a simple method to enhance the emission, biocompatibility and cellular permeability of common chromophores, which may open more avenues for the applications of originally non- or poor fluorescent PAHs.

[1] School of Materials Science and Engineering, Nanyang Technological University, Singapore 639798, Singapore. [2] Ningbo Institute of Materials Technology and Engineering, Chinese Academy of Sciences, No. 1219 ZhongGuan West Road, 315201 Ningbo, China. [3] Department of Chemistry, National University of Singapore, 3 Science Drive, Singapore 117543, Singapore. [4] State Key Laboratory of Applied Organic Chemistry, Lanzhou University, Tianshui Southern Road 222, Lanzhou 730000 Gansu Province, China. Correspondence and requests for materials should be addressed to H.-L.Z. (email: Haoli.zhang@lzu.edu.cn) or to Q.Z. (email: qczhang@ntu.edu.sg)

Organic semiconductors, especially π-conjugated metal-free luminescent organic materials, are crucial for human beings to deliver and illuminate information[1–3] via modern technologies (e.g., photonics, optoelectronic devices, biological probes, and fluorescent sensors[4–9]) due to their flexibility, brightness, easy-processability, tunable photophysical properties[10–12], high optical gain[13], and possible biocompatibility[14]. In general, most of the above-mentioned applications require solid or aggregate-state luminescent materials, such as thin films, bulk crystals, or nanoparticles[15]. However, the features of planar aromatic structures of conventional organic chromophores, which facilitate the π–π stacking of the chromophores and other physical interactions (e.g., energy transfer, inter-, or intramolecular charge transfer, and excited states reactions)[16–18], further makes these materials suffer from the well-known aggregation caused quenching (ACQ) effect[4–7,19–22]. Clearly, such ACQ effect has greatly impeded their practical applications in optoelectronic devices and biological applications[1–3,23]. Thus, developing strategies to address ACQ effects are of utmost importance. In the past decade, although lots of methods in various fields have been adopted to reduce ACQ effects[4–9,19–28], only a few approaches are successful due to the inherent strong tendency of π–π stacking of fluorescent molecules[21,22,24]. These examples include, controlling the molecular stacking to form J-aggregates, decreasing the crystalline degree via introducing amorphous domains, avoiding a slow intersystem crossing (ISC) process and controlling triplet formation[25–28], and attaching long alkyl chain substitution onto chromophores to prevent π–π interaction between conjugated planes in aggregate states[29]. Very recently, a variety of fluorescent materials with famous aggregation-induced emission (AIE) properties have been prepared and widely used in many fields[4–7,29–33]. However, there are still severe problems remaining in above-mentioned methods. For example, the molecular stacking was difficult to be controlled[34–38]. The method to avoid slow ISC process can result in retaining significant losses of the residual triplet states[39]. Furthermore, some AIE molecules are difficult to be synthesized and purified, leading to a higher cost[21,22,39,40]. Thus, developing facile strategies to overcome above-mentioned issues is highly desirable.

In our previous research, we have already found that the cocrystals between octafluoronaphthalene (OFN with high-energy gap, photoluminescence quantum yield (PLQY) in solid is 2.6%) and planar coronene and anthracene (Cor and Ant for short, with low-energy gaps and ACQ effect) exhibited enhanced PLQY[39]. This result further encourages us to investigate more polycyclic aromatic hydrocarbon (PAH) chromophores (perylene and pyrene, Per and Pyr for short) because most of them have low PLQYs in solid (known as ACQ effect). Moreover, it is very important to reduce the sizes of fluorescent cocrystals to micro/nanoscales, because such nano/micromaterials can have huge potential applications in optoelectronic fields and bioimaging due to their good biocompatibility, easily surface modification with functional molecules[41–44], and readily accumulating in most disease regions[45–51].

Herein, we demonstrate a facile method (Fig. 1) to regularly co-assemble two different molecules into well-ordered aggregates[52–54], where weakly-fluorescent OFN is used as molecular barrier, Cor and Per were chosen as PAHs chromophores, and P123 (poly(ethylene glycol) (PEO$_{20}$–PPO$_{70}$–PEO$_{20}$))[55] is selected as a biocompatible surfactant to protect the micro/nanostructures from further growth in aqueous conditions. Significantly, the as-fabricated crystals can be easily controlled in the range of micro (gram-scale yield) or nanoscale via changing the concentrations of P123 or chromophores. Notably, the solid PLQYs for as-obtained four micro/nanococrystals are enhanced by 254%, 235%, 474%, and 582%, respectively. The "NTU" patterns written from Cor/OFN MW and Per/OFN MW powders clearly exhibit much stronger emission than single-component ones (Cor or Per), which further confirms the greatly improved solid fluorescence of the PAHs. Protected by hydrophilic PEO chains, the resultant cocrystals cannot aggregate together in solution and were endowed with superb dispersibility in water at a high concentration (10 mg mL$^{-1}$). More importantly, the superb water dispersibility and highly improved PLQY make the as-prepared nanococrystals present good biocompatibility and cellular permeability, less cytotoxicity, and considerable cell imaging performance. This work indicates that the emission, biocompatibility and cellular permeability of common chromophores (PAHs with ACQ effect in aggregated state) can be greatly enhanced via such facial co-assembly method, in which OFN was used as a molecular barrier. This research can open more avenues for the applications of originally non- or poor-fluorescent PAHs.

## Results

**Preparation of micro and nanococrystals**. The preparation processes of micro/nanococrystals are illustrated in Fig. 1. As shown in Fig. 1a, the micro/nanocrystals (Per MS, Per/OFN MW, Cor MW, and Cor/OFN MW) were prepared as following: 200 mL of PAH/OFN tetrahydrofuran (THF) solution (a total constant concentration of 10 mg mL$^{-1}$) with certain mole ratios of PAH to OFN (1:0, 1:0.5, 1:1, 1:2, 1:5, 1:8, and 1:10) was rapidly added into 800 mL of as-prepared 0.125 mM P123 (a common biocompatible surfactant, $M_n$ ≈5800) aqueous solution, and then, the as-prepared solutions were allowed to stand for 12 h. The resultant cocrystals can be easily collected via filtration, which gave gram-scale products (~1.95 g, 97%). After optimization, the highest PLQYs for both Per/OFN and Cor/OFN cocrystals appeared at 1:1 PAH/OFN (See Supplementary Table 1 and Supplementary Fig. 1). Therefore, all the following studies based on PAH/OFN ratios were 1:1 and 1:0 (only PAH, for control). Nanococrystals, such as Per NS, Per/OFN NP, Cor NR and Cor/OFN NR (Fig. 1b), were fabricated through changing the concentration of chromophore/OFN solution and P123 solution to 1 mg mL$^{-1}$ and 1.25 mM, respectively. Also, the as-resulted solutions were kept sonicating for 30 min before harvesting nanococrystals.

Tables 1, 2 show the key information of the as-obtained Per MS, Per/OFN MW, Per NS, Per/OFN NP, Cor MW, Cor/OFN MW, Cor NR, and Cor/OFN NR with different uniform assemblies, where MW, NR, MS, NS, and NP are defined as microwire, nanorod, microsheet, nanosheet, and nanoparticle, respectively. As shown in Tables 1, 2, and Supplementary Table 1, all solid PLQYs of the as-prepared micro/nanocrystals have been enhanced. Especially for Cor/OFN and Per/OFN nanococrystals, their PLQYs can be improved by 582% and 474%, respectively. The reason is that OFN can greatly hinder intermolecular interaction and electron exchange between the chromophores (namely, blocking the ISC from singlet state to triplet state)[56,57].

**Characterization of micro and nanococrystals**. The successful doping of OFN in Per/OFN and Cor/OFN cocrystals were supported by Fourier Transform Infrared (FTIR) spectra, Raman spectra, differential scanning calorimetry (DSC) curves, and powder X-ray diffraction (PXRD) results. The characteristic FTIR peak at 788 cm$^{-1}$ [58] of C–F bond in OFN moved to 782 cm$^{-1}$ for Per/OFN MW (see Supplementary Fig. 2) and 780 cm$^{-1}$ for Cor/OFN MW (see Supplementary Fig. 3), respectively. Moreover, because of being surrounded by electron-rich PAHs, the stretching peaks of OFN at 1202 cm$^{-1}$ shifted to 1194 cm$^{-1}$ (see Supplementary Figs. 2 and 3). Such results suggest that the C–F bond was weakened and lengthened as a result of $n \rightarrow \sigma^*$ donation

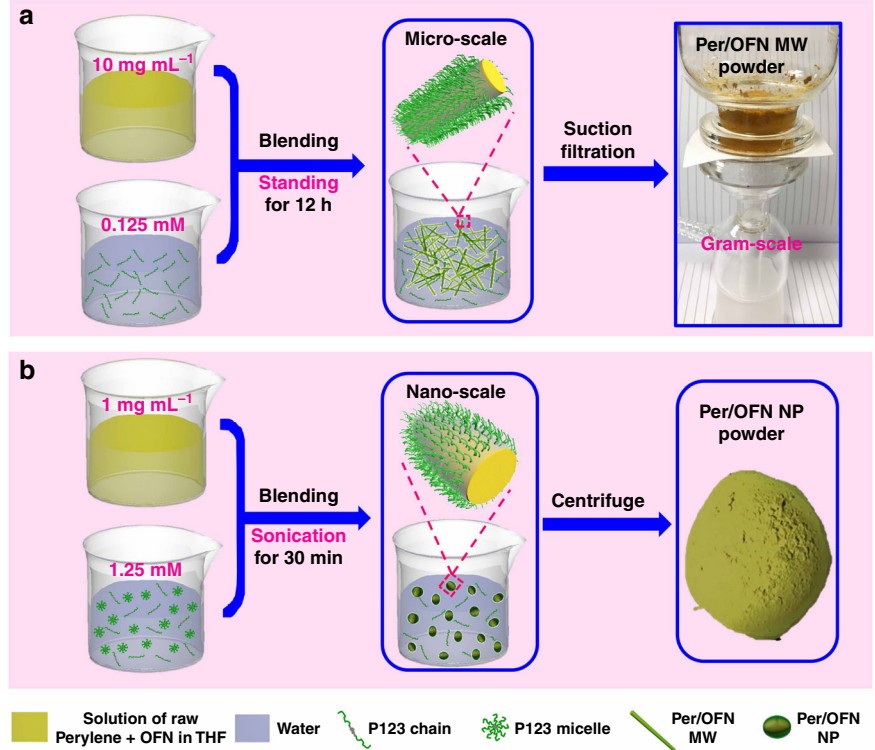

**Fig. 1** Illustrations of preparation processes. Morphologies (middle) and powder pictures (right side) for the micro/nanococrystals (taking Per/OFN cocrystals as representative here, other cocrystals were prepared via the same procedures). **a** Preparation procedure of Per/OFN microwires (Per/OFN MW), gram-scale cocrystals (ca. 1.95 g) can be obtained from 200 mL of THF solution (10 mg mL$^{-1}$) of chromophore/OFN, concentration of P123 is 0.125 mM. **b** Preparation procedure of Per/OFN nanoparticles (Per/OFN NP) via changing the concentrations of Per (1 mg mL$^{-1}$) and P123 (1.25 mM). We created the images of beakers by ourselves via 3D max software 2014

### Table 1 Key information of the resultant micro/nanocrystals

| Sample name | Raw Cor[a] | Cor MW | Cor/OFN MW | Cor NR | Cor/OFN NR |
|---|---|---|---|---|---|
| Mole ratio [PAH/OFN] | – | 1:0 | 1:1 | 1:0 | 1:1 |
| Morphology | – | Microwire | Microwire | Nanorod | Nanorod |
| Length (L) | – | 20–60 µm | 10–60 µm | 100–500 nm | 50–500 nm |
| Thickness (T)/diameter (D)[b] | – | 0.2–1 µm | 0.1–0.5 µm | ~100 nm | ~100 nm |
| PLQY [%][c] | 3.4 | 5.8 | 11.4 | 12.8 | 23.2 |
| Enhancement [PLQY, %][d] | – | 71% | 235% | 276% | 582% |

[a]Coronene as received
[b]Thickness for sheets, diameter for wire, rod and particle, which were obtained from TEM and AFM images
[c]Absolute quantum yield tested from integrating sphere method. [d] Enhancement of the PLQY of as-prepared cocrystals compared with raw PAHs, the PLQYs are the average values of three tests

### Table 2 Key information of the resultant micro/nanocrystals

| Sample name | Raw Per[a] | Per MS | Per/OFN MW | Per NS | Per/OFN NP |
|---|---|---|---|---|---|
| Mole ratio [PAH/OFN] | – | 1:0 | 1:1 | 1:0 | 1:1 |
| Morphology | – | Microsheet | Microwire | Nanosheet | Nanoparticle |
| Length (L) | – | 2–3 µm | 20–80 µm | 100–300 nm | – |
| Thickness (T)/diameter(D)[b] | – | 37 nm | 0.4–2 µm | 17 nm | 100–150 nm |
| PLQY [%][c] | 4.6 | 7.9 | 16.3 | 17.2 | 26.4 |
| Enhancement [PLQY, %][d] | – | 72% | 254% | 274% | 474% |

[a]Perylene as received
[b]Thickness for sheets, diameter for wire, rod and particle, which were obtained from TEM and AFM images
[c]Absolute quantum yield tested from integrating sphere method. [d] Enhancement of the PLQY of as-prepared cocrystals compared with raw PAHs, the PLQYs are the average values of three tests

in cocrystals[59]. Additionally, the peak at 947 cm$^{-1}$ (C–F side-to-side stretch) of OFN migrated to 942 cm$^{-1}$ for Per/OFN MW (see Supplementary Fig. 2) and 943 cm$^{-1}$ (see Supplementary Fig. 3) for Cor/OFN MW, respectively[58]. Conversely, the strong absorption peaks at 3048 cm$^{-1}$ (stretching of Ar–H in Per) and 3046 cm$^{-1}$ (the stretching of Ar–H in Cor) moved to 3067 cm$^{-1}$ (see Supplementary Fig. 2) and 3057 cm$^{-1}$ (see Supplementary Fig. 3), respectively. These changes are ascribed to C–F⋯H interactions and demonstrated the succeful packing between electron-rich Per and electron-deficient OFN[59]. The same results were also obtained in Raman spectra. The peaks at 358, 976, and 3061 cm$^{-1}$ in Raman spectrum of Per MS were blue-shifted to 363, 978, and 3067 cm$^{-1}$, respectively (see Supplementary Fig. 4). Likewise, after stacking with OFN, the peaks at 1025 and 3040 cm$^{-1}$ from Cor MW moved to 1027 and 3057 cm$^{-1}$, respectively (see Supplementary Fig. 5). These results indicated the decreased electron density of Per and Cor molecules in cocrystals[60]. Furthermore, the temperature at the single sharp peak in the DSC curve corresponds to the melting point. Evidently, after being doped with OFN, the melting point of Per/OFN MW increased to 355 °C, which was different from individual OFN crystals or Per MS (see Supplementary Fig. 6). But for Cor MW and Cor/OFN MW, they had already decomposed (indicated by the much broader peaks) before the melting points appearred due to unstability and high melting point of Cor (see Supplementary Fig. 7). The DSC results indicated good thermostability of the cocrystals due to the strong intermolecular interaction. In order to further confirm the crystal structures, PXRD curves were collected. The XRD patterns of Per/OFN and Cor/OFN cocrystals were different from that of constituent OFN or PAH molecules (see Supplementary Figs. 8 and 9), which further confirmed the successful formation of the cocrystals.

**Crystal structure analysis**. To confirm the detailed molecular stacking in the resultant Per/OFN and Cor/OFN cocrystals, the experimental PXRD results were compared to the calculated ones from crystal structures of the corresponding bulk single crystals. As expected, the experimental PXRD result of Per/OFN was almost the same as the simulated one calculated from Per/OFN single cocrystal structure (See Supplementary Fig. 10), which suggests the same molecular stacking in the micrococrystals as that in Per/OFN single cocrystals. A mixed-stack mode was observed in the stacking structure, where Per and OFN in the cocrystals stacked on each other alternately via face-to-face stacking along [100] direction with a mean distance of 3.440 Å between Per-OFN and 6.886 Å between Per–Per (see Supplementary Fig. 11). Similarly, for Cor/OFN MW, the experimental peaks indicated the same locations as the simulated peaks of the Cor/OFN-based single cocrystals, which elucidated the same molecular stacking mode in Cor/OFN MW as that in Cor/OFN single cocrystals[39]. Specifically, two component molecules in the cocrystals took a mixed-packing mode and stacked on each other alternately (see Supplementary Fig. 12), which adopted a face-to-face stacking pattern along [100] direction with a mean distance of 3.440 Å between Cor–OFN and 6.872 Å between Cor–Cor (see Supplementary Fig. 13), respectively. In addition, as indicated from the simulation based on the minimum energy principle using Material Studio, the [100] and [001] directions (π–π stacking direction) are the preferring growth directions for both Per/OFN and Cor/OFN, which can lead to the formation of 1D wires. The morphologies were visualized by electron microscopes and will be detailed below.

**Visualization and analysis of morphology**. For the construction of microassemblies (microcrystals), a higher PAH concentration

(10 mg mL$^{-1}$) and a lower P123 concentration (0.125 mM) were used here. Once hydrophobic PAH or PAH/OFN were added, these molecules simultaneously began to assemble together under several driving forces such as π–π stacking, F⋯H interaction (for cocrystals), and hydrophobic effect, until the system reached equilibrium. Eventually, Per MS (Fig. 2a), Per/OFN MW (Fig. 2b), Cor MW (Fig. 2e), and Cor/OFN MW (Fig. 2f) formed, which were all-water dispersible due to the protection of P123 from further aggregation. In addition, the dimensions of these assemblies were stable and no longer changed after standing for 12 h. As shown in Fig. 2a, Per could be assembled into regular two-dimensional (2D) microsquare-sheets. Statistics gave a length (L) of 2–3 μm (Fig. 2a and see Supplementary Fig. 15a). AFM profile manifested one microsquare sheet with an extremely uniform thickness of ∼37 nm (inset of Fig. 2a). As expected, after being co-assembled with OFN, the resultant Per/OFN aggregates (Fig. 2b) were totally different from that of Per MS (Fig. 2a and see Supplementary Fig. 15a) and OFN (see Supplementary Fig. 14) crystals and presented one-dimensional (1D) microwires with the length of 20–40 μm and the diameter of 0.4–2 μm (Fig. 2b and See Supplementary Fig. 15b, c). The morphology of the 1D wire was also visualized via AFM (inset in Fig. 2b) and transmission electron microscopy (TEM) images (Fig. 2c and see Supplementary Fig. 16). Differently, single-component Cor with much larger planes tended to align to 1D microwires with the length of 20–60 μm and the diameter of 0.2–1 μm from the statistics based on all the assembles in the scanning electron microscope (SEM) image of Cor MW (Fig. 2e and see Supplementary Fig. 17a–c). The AFM profile (inset of Fig. 2e) magnified one wire, which demonstrated the same morphology as that shown in SEM image. However, when regularly stacked with OFN molecules, the Cor/OFN MW crystals became much more uniform for their diameter (0.1–0.5 μm) and much more rigid (Fig. 2f, g, see Supplementary Fig. 17d, e) than that of Cor MW, which were also completely different from OFN crystals (see Supplementary Fig. 14). An attempt was made to further confirm the crystal structures of two cocrystals by selected area electron diffraction (SAED) and crystal lattice. However, no crystal lattice (see Supplementary Figs. 16c and 18c) and diffraction patterns (see Supplementary Figs. 16d and 18d) were observed, which might be attributed to the highly energetic irradiation of electron beams that can destroy crystal structures[61]. Furthermore, Fig. 2d, h presented the predicted growth morphologies of Per/OFN and Cor/OFN cocrystals, respectively, based on the calculated energies (see Supplementary Tables 3 and S4, respectively), which are similar to the experimental results.

**Optical properties**. Being co-stacked with electron-deficient OFN molecules, which possess a higher band gap of ca. 3.78 eV[39], the photophysical properties of the micrococrystals have been changed, as shown in Fig. 3. The absorption peak of Per/OFN MW was blue-shifted by 15 nm according to the location of the rightmost peaks compared with the pure Per MS crystals (Fig. 3a). The luminescence properties of the as-prepared micrococrystals were preliminarily investigated via photoluminescence spectroscopy (Fig. 3b, c, e, f). As shown in Fig. 3b, the main peak of the PL spectra of Per MS centered at ∼583 nm, and that of OFN crystals appeared at 351 nm. Interestingly, when assembled together to form Per/OFN MW, a blue-shifted emission (shifted by 29 nm) centered at 554 nm as well as two new peaks appeared. The absorption and emission spectra of the crystals based on Cor were given in Fig. 3d–f. Similarly, the absorption and emission peaks of Cor/OFN MW were blue-shifted by 9 and 16 nm, respectively, comparing to that of the Cor MW crystals. The blue-shifted phenomena in the absorptions of Per/OFN MS and Cor/OFN

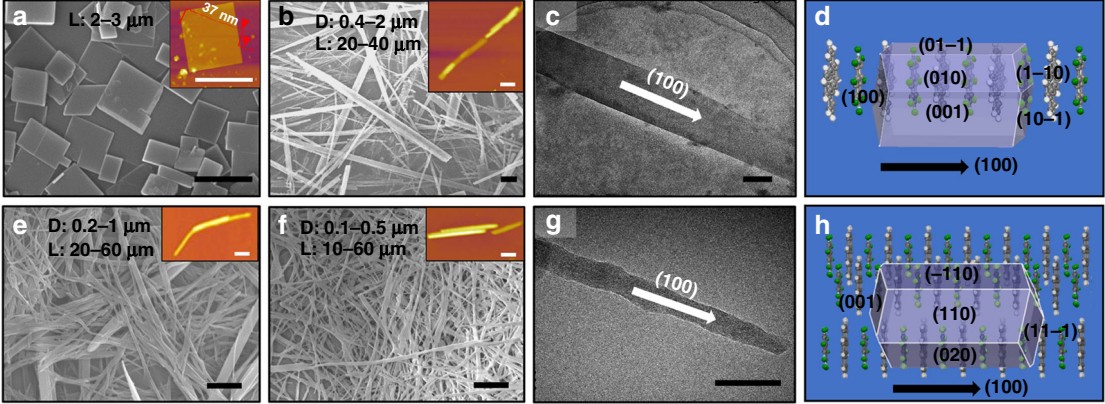

**Fig. 2** Morphologies characterizations of the micro/nanocrystals. Per MS (**a**), Per/OFN MW (**b, c, d**), Cor MW (**e**), and Cor/OFN MW (**f, g, h**). **a, b, e, f** Low-magnification SEM images of the four crystals, in which D and L denote the diameter and length, respectively. The insets are the corresponding AFM images, the height profile is given in inset of (**a**). **c, g** Low-magnification TEM images. **d, h** The predicted growth morphologies of the Per/OFN and Cor/OFN cocrystals based on the calculated energies. The scale bars in **a, b, e, f** are 3 μm, the scale bars in **c, g** are 500 nm

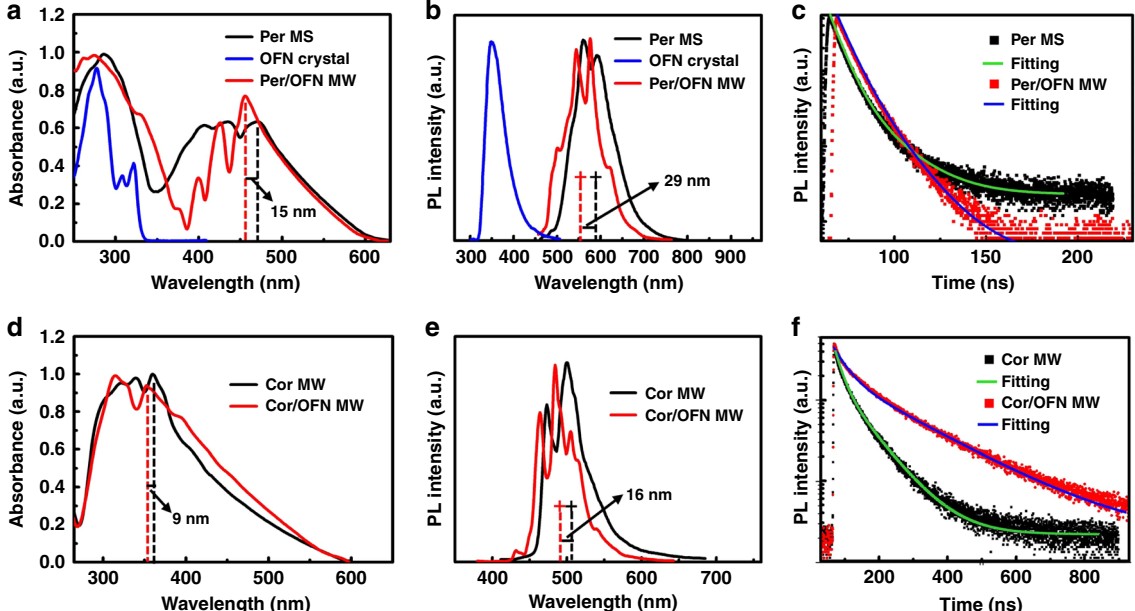

**Fig. 3** Photophysical properties of the microcrystals in solid states. (**a**) Absorption spectra, (**b**) PL spectra and (**c**) Time-resolved PL measurements of Per MS and Per/OFN MW under the excitation of 460 nm. (**d**) Absorption spectra, (**e**) PL spectra and (**f**) Time-resolved PL measurements of Cor MW and Cor/OFN MW under the excitation of 360 nm. All the curves were normalized to make the maximum intensities to be a same value for the purpose to compare their peak positions

MW were attributed to both the electron-deficient periodic OFN molecular barriers that intercalated between Per–Per and Cor–Cor via π–π interaction and blocking the formation of Per–Per and Cor–Cor dimers, which was further proven by time-dependent density functional theory calculations[39]. The PL blue-shifts are attributed to the screening of the π-interactions between PAH and PAH by OFN molecules, which can evidently decrease the exciton-delocalization-induced PL red-shift[62]. In addition, the PL decays of micro-Per/OFN cocrystals under an excitation of 460 nm gave a similar PL lifetime (average lifetime, τ ~10 ns) as that (~8 ns) of the Per pristine crystal (Fig. 3c) (for detailed calculation refer to Page S19). Likewise, the Cor/OFN MW showed a longer PL lifetime (~160 ns) than that (58 ns) of single-component Cor MW (Fig. 3f).

**Fluorescent study**. Surprisingly, the intercalation of OFN not only led to the changes in above-mentioned evidence, but also

significantly improved the PLQYs of as-prepared cocrystals. The cocrystals presented much higher PLQYs than that of the PAHs in pristine crystals even in solution for Per and Cor (Tables 1 and 2, see Supplementary Table 1). Specifically, the PLQYs of Per/OFN MW and Per/OFN NP increased by 254% and 474% (Table 1), respectively, in comparison to the PLQY (4.6 %) of raw Per powder as received, which even were improved by 96% and 218%, compared with the corresponding Per solutions in chloroform (PLQY, 8.3%). While for Cor/OFN MW and Cor/OFN NR, the PLQYs were enhanced by 235% and 582%, respectively, compared to the PLQY (3.4%) of raw Cor powder, which has a low PLQY of 4.6% in solution[39]. As shown in Fig. 4c, the introduction of colorless OFN did not cause the color change of PAHs, instead, just resulting in a color-diluted phenomenon. Particularly, as expected, when 365 UV lamp was used, the cocrystal powders as well as the "NTU" patterns demonstrated much brighter luminescence (yellow and blue for Per/OFN MS

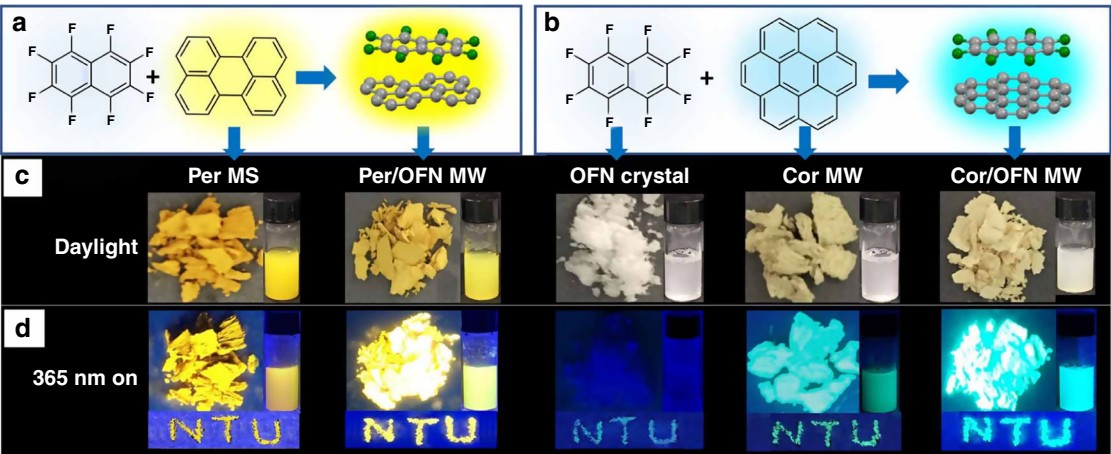

**Fig. 4** cocrystallization schemes of Per/OFN MW (**a**) and Cor/OFN MW (**b**) and corresponding photos. The corresponding photographs of the five as-prepared powders (OFN crystals, Per MS, Per/OFN MW, Cor MW, and Cor/OFN MW) and the patterned "NTU" as well as their aqueous dispersions (can be considered as all-water dispersed fluorescent inks) under daylight (**c**) and 365 nm UV lamp on (**d**)

and Cor/OFN MW, respectively) in contrast to that of each corresponding pure PAH powders and OFN crystals (Fig. 4d). To verify the universality of this method, anthracene (Ant) and pyrene (Pyr), which have intrinsically higher PLQYs, were also chosen for further investigation. The experimental results indicated that the fluorescence of Ant and Pyr were also improved, and their PLQYs were increased by 97% and 18%, respectively, which further demonstrated the universality of the method presented in this work. Note that such bright luminescence can be definitely comparable to the common dyes[1–9,19–28], which could be potentially employed as a kind of highly emissive solid materials obtained just via doping OFN (using a facial method with high yield and efficiency) into commercially-available PAH chromophores (normally possess low solid PLQYs and can be easily categorized as nonemissive solid materials). Another exciting factor is that, in addition to obtaining the solid luminescence, the water dispersibility of such materials were also realized (insets in Fig. 4c, d). The as-resulted homogeneous dispersions demonstrated the same brighter luminescence as the solid ones and were stable for at least six months for nanoscale crystals (Per/OFN NP and Cor/OFN NR). The reason for the remarkably increased PLQYs was given in our previous work[39]. Specifically, weakly fluorescent OFN with a higher band gap of ~3.78 eV was used as the molecular barrier to intercalate in the periodic packed structures of PAHs, which can hinder intermolecular interaction and electron exchange between the planar molecules such as Per–Per or Cor–Cor[56,57], whose band gaps are ~2.76 and ~3.10 eV[63,64], respectively, eventually leading to the absence of triplet states, as demonstrated in the schematic (see Supplementary Fig. 19)[39]. In addition, the PLQY enhancements resulted from the reduced dimensions using P123 as the surfactant also can be understood because P123 can effectively block the π–π interaction between the free PAH molecules and as-formed small crystals[29], which not only decreases the dimensions of the resultant crystals but also weakens the ACQ effect to a certain extent, and therefore particularly improved the PLQYs (Tables 1, 2).

**Cell imaging**. Importantly, the excellent water dispersibilities of Per/OFN NP and Cor/OFN NR offer opportunities for the exploration of their applications in bio-systems. As a proof of such idea, we examined cell imaging behaviors of these crystals. Given that cells cannot eat too large foreign objects, the assemblies were controlled to nanoscale, which were realized under the guidance of surfactant templates (micelles with a hydrophobic cores and

hydrophilic shells) formed in a higher concentration (Fig.1b), and the PAH molecules can either intercalate or attach into the micelles via hydrophobic effect, eventually realized the co-assembly[65]. Remarkably, nanoscale Per square sheets (length, 100–300 nm; thickness, ~17 nm), Per/OFN particles (diameter, 100–150 nm), Cor rods (length, 100–500 nm; diameter, ~100 nm) and Cor/OFN rods (length, 50–500 nm; diameter, ~100 nm) were successfully prepared via above conditions (Tables 1, 2, Fig. 5a, d, g, j and Figs. S23, 24). Protected by hydrophilic PEO chains from P123, the as-prepared cocrystals were endowed with superb dispersibility in water with high concentrations (10 mg mL$^{-1}$). Note that the dispersions (1 mg mL$^{-1}$) can be stable without any visual precipitation at least for 6 months (see Supplementary Fig. 25) and the stability of these dispersions was further confirmed by the nearly unchanged morphologies (see Supplementary Fig. 26), PL spectra, and PLQYs (see Supplementary Fig. 29). Benefiting from the highly PL emissive properties and expected biocompatibility, the as-prepared nanocrystals might be employed to explore their cell imaging properties. However, one requirement is that these particles should have noncytotoxicity to cells. To address this issue, the vitro cytotoxicity of four nanocrystals has been evaluated via the standard MTT (3-(4,5-dimethylthiazol-2-yl)-2,5-diphenyltetrazolium bromide) assay using MCF-7 cells (see Supplementary Fig. 30). As expected, low cytotoxicity of such nanocrystals has been confirmed. Specifically, over 90% cell viability was observed after incubating MCF-7 cells with any of four crystals with different concentrations ranging from 0 to 50 μg mL$^{-1}$, which can be attributed to the protection of a large amount of biocompatible PEO chains. Based on above cytotoxicity results, the concentration of 30 μg mL$^{-1}$ of the crystals was used for further imaging experiments. After a 4-h incubation of MCF-7 cells with Per NS, Per/OFN NP, Cor NR, and Cor/OFN NR, respectively, the live cells were visualized through confocal microscope under a single laser excitation of 405 nm. As shown in the confocal micrographs (Fig. 5b, e, h, k), almost no emissions presented in the cells, which have no any nanocrystals to incubate with. Once incubated with Per/OFN NP and Cor/OFN NR, the live cells displayed highly bright emissions with no any changes in their morphologies (Fig. 5f, l). Also, the live cells incubated with Per/OFN NP and Cor/OFN NR were much brighter than the cells incubated with no OFN-doped nanocrystals (Fig. 5c, i). Furthermore, the emissions located at cytoplasmic regions suggested that all the fluorescent nanocrystals can penetrate through cell membranes successfully, which benefited from their hydrophilic and biocompatible surface.

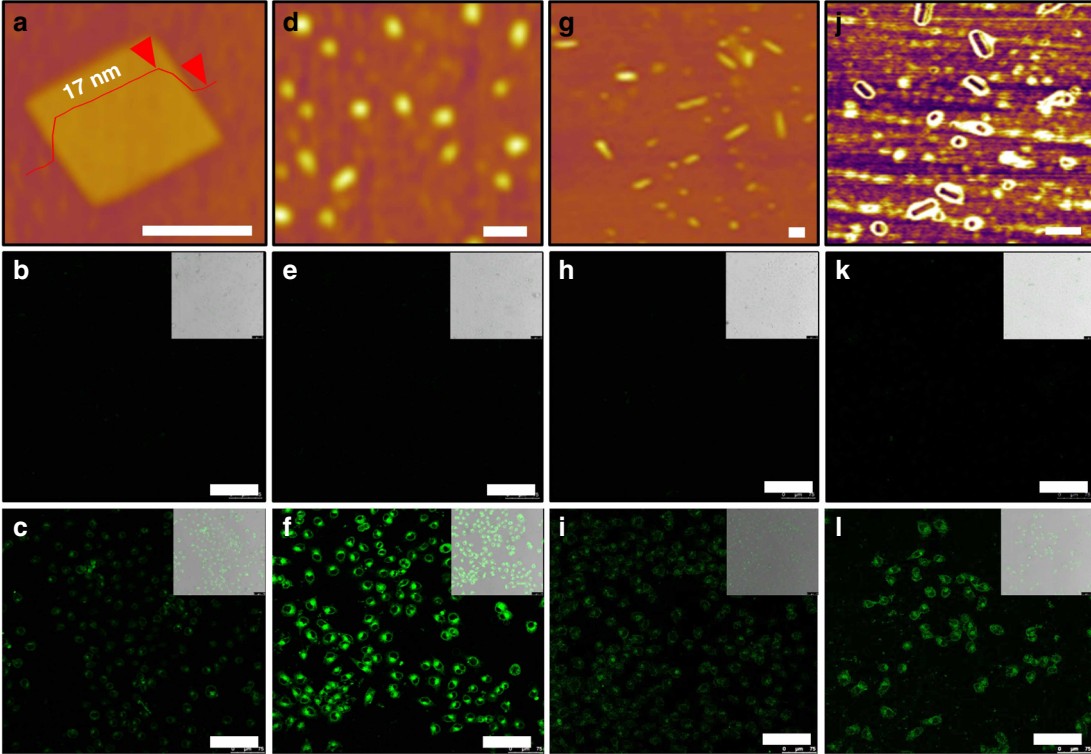

**Fig. 5** Cell imaging results. AFM height/phase images of nanoscale Per NS (**a**), Per/OFN NP (**d**), Cor NR (**g**), and Cor/OFN NR (**j**). Confocal images of MCF-7 cells and labeled by Per NS (**c**), Per/OFN NP (**f**), Cor NR (**i**), and Cor/OFN NR (**l**), as well as the corresponding controls unlabeled by any samples (**b**, **e**, **h**, **k**). All the insets are the results under bright field. $\lambda_{ex} = 405$ nm. Scale bars in **a**, **d**, **g**, **j** are 200 nm. Scale bars in **b**, **c**, **e**, **f**, **h**, **i**, **k**, **l** are 75 μm

## Discussion

We have greatly improved the PLQYs of the micro/nanocrystals of Per/OFN MW, Cor/OFN MW, Per/OFN NP, and Cor/OFN NR via such facile, low cost and efficient co-assembly method. Here we use weakly fluorescent OFN with higher band gap as a molecular barrier, which can intercalate in the periodic packed structures of Per and Cor, and then greatly hinder intermolecular interaction and electron exchange between the planar PAHs, eventually leading to the absence of triplet states. We further verified the universality of this method via choosing Ant and Pyr as PAH chromophores, even they have intrinsically higher PLQYs. Remarkably, the enhanced bright luminescence of the cocrystals based on commercially available PAH chromophores (poor luminescence in solid) and OFN can be definitely comparable to that of the common dyes, which could make them to be potentially employed as a kind of highly emissive solid materials. Meanwhile, the resultant cocrystals are endowed with superb dispersibility in water via using a biocompatible P123. Profiting from the superb water dispersibility and highly emissive nature, Per/OFN NP and Cor/OFN NR presented good biocompatibility and cellular permeability, less cytotoxicity, and good cell imaging results. This research will provide constructive guidelines for fabricating highly solid emissive materials by using traditional ACQ molecules (PAHs) as raw materials, which may greatly expand the family of solid emissive materials. More importantly, it opens more avenues for the applications of originally non- or poor fluorescent PAHs (ACQ effect in aggregated state) in bioimaging or optical devices.

## Methods

**Materials.** All the materials were purchased from Sigma-Aldrich suppliers and used as received, unless otherwise noted. Octafluoronaphthalene (OFN, 96%), perylene (Per, 99%), coronene (Cor, 97%), pyrene (Pyr, 98%), and anthracene (Ant, 99%) were purchased from Sigma-Aldrich. THF (99%) was purchased from Merck,

Amphiphilic triblock copolymers poly(ethylene glycol)$_x$-block-poly(propylene glycol)$_y$-block-poly(ethylene glycol)$_x$ (P123, $x = 20$, $y = 70$, average $M_n \approx 5800$) was purchased from Polymer Source Inc. 200 mesh copper specimen grids (ZB-C4000) with ultra-thin carbon support film (thickness ≤ 3 nm) were purchased from Beijing Zhongxing Bairui Technology Co., Ltd. A silicon wafer (about 0.25 cm²) was purchased from Shanghai Xinyang Semiconductor Materials Co., Ltd. and pretreated with O² plasma for 15 min to make it more hydrophilic. Deionized water (DI water, resistance > 18.2 MΩ cm⁻¹) was used for all experiments.

**Characterization methods.** Fourier transform infrared (FTIR) spectra were tested on a Spectrum Frontier FTIR spectrometer (Perkin Elmer, Inc., USA). Raman spectroscopy was performed on the Nicolet iS50 FTIR spectrometer equipped with the iS50 Raman module. PXRD patterns were recorded on a Bruker D8 diffractometer (German) equipped with Cu Kα radiation ($\lambda = 1.5406$ Å) at a scan rate of 0.02 deg s⁻¹. Single-crystal XRD data collection of Per/OFN and Cor/OFN crystals were performed on a Bruker SMART APEX-II CCD area detector equipped with a D8 goniometer. All data were collected using graphite-monochromated and 0.5 mm-Mono Cap-collimated Mo-Kα radiation ($\lambda = 0.71073$ Å) with the ω-scan method. All data were processed with the SAINT program of the APEX3 software for reduction and cell refinement. Multiscan absorption corrections were applied by using the SADABS program for area detector. All structures were solved by the direct method and refined by the full-matrix least-squares method on $F^2$ (SHELX-2014). All non-H atoms were refined anisotropically. Hydrogen atoms were placed in idealized positions and included as riding with $U_{iso}$ (H) = 1.2 $U_{eq}$ (C). Crystallographic data and structural refinements are summarized in Supplementary Table 2. The fluorescence quantum yield (Φ) was determined by a quantum efficiency measurement system: QE-2100 (Qtsuka Electronics Co. Ltd., Japan) at 25 °C. Time-resolved fluorescence: time-resolved fluorescence measurements were made on a Fluoromax-4 Spectrofluorometer (France) at 25 °C. DSC curves were recorded on a DSC TA instruments Q10 (USA) under flowing (50 mL min⁻¹) nitrogen, all the tests were conducted at an increasing rate of 5 °C min⁻¹. Ultraviolet–visible (UV–vis) absorption spectra were obtained on UV–vis–NIR Cary 5000 spectrophotometer. Fluorescence spectra were collected via a Cary Eclipse Fluorescence Spectrophotometer. SEM observations were carried out on a JEOL/JSM-6340F at an electric voltage of 5 kV. The samples were prepared via dip-coating method using the dispersions (1 mg mL⁻¹) of the micro/nanocrystals in water and dried under room temperature (RT) in the air for 12 h before test. Atomic force microscope (AFM) observations were performed on a scanning probe microscope (AFM Asylum Research Cypher S, USA) operated in AC Mode Imaging in Air by using Arrow-NCR-50-Silicon SPM-Sensor (Coating: detector, side: Al-coating) cantilevers with a force constant of 42 N m⁻¹.

AFM samples were prepared via dip-coating method using the dispersions ($1 \times 10^{-3}$ mg mL$^{-1}$) of micro/nanocrystals in water, and dried under RT in the air for 12 h before test. TEM images were obtained on a JEM-2100 (JEOL Ltd., Japan) with accelerating voltage of 200 kV. TEM samples were prepared by dropping the crystal dispersions onto carbon-coated copper grids, followed by air-drying for 12 h. No staining treatment was performed. Ultrasonic cleaner (DKSONIC, DK-1000D, 500 W) was used to prepare nanoparticles under RT. All the photos were taken by a Cannon EOS 700D camera under the irradiation of hand-held UV lamp light (on and off).

**Preparation of micrococrystals.** Firstly, 0.125 mM poly(ethylene glycol)-block-poly(propylene glycol)-block-poly(ethylene glycol) (P123, $M_n \approx 5800$, CMC = 0.313 mM at 20 °C[4]) aqueous solution was prepared 12 h in advance to ensure that P123 can be dissolved completely. Secondly, the chromophore/OFN solutions (at constant total concentration of 10 mg mL$^{-1}$) with different chromophore to OFN mole ratio (1:0, 1:0.5, 1:1, 1:2, 1:5, 1:8, and 1:10) were prepared by dissolving chromophores and OFN into THF. Afterward, 20 mL chromophore/OFN solution was added into 80 mL above P123 aqueous solution and mixed rapidly to form uniform emulsion, which was then allowed to stand for 12 h. After that, the resulted cocrystals were collected via filtration under vacuum and were washed with water for three times to remove the free P123 chains. Followed by air-dry, about 195 mg of water-dispersible micrococrystals can be obtained, which gave a yield of ~ 97%. In addition, gram-scale cocrystals also can be easily realized by using much more above chromophore/OFN solution as well as P123 aqueous solution. For example, when 200 mL chromophore/OFN solution was added into 800 mL above P123 aqueous solution, in this case, ~1.95 g products can be easily obtained.

**Preparation of nanococrystals.** The dimensions of cocrystals can be controlled via decreasing the concentration of chromophore/OFN solution as well as increasing the content of P123, which can meet the application requirements of more fields. For example, in order to make the fluorescent cocrystals be suitable for bioimaging, the dimensions should be controlled to be nanoscale, which were prepared as follow. Aqueous solution of P123 with a higher concentration of 1.25 mM was prepared 12 h in advance firstly. Second, the chromophore/OFN solutions (total concentration, 1 mg mL$^{-1}$) with chromophore to OFN mole ratio of 1:1 were prepared by dissolving chromophores and OFN into THF. Then 20 mL chromophore/OFN solution was added into 80 mL above P123 aqueous solution under continuous ultrasonication rapidly, the mixture was sonicated for another 30 min, which was then allowed to stand for 12 h. Afterward, the resulted nanococrystals were collected via centrifugation procedures: centrifuged under 9500 r min$^{-1}$ for 5 min, then removed the liquid of upper layer, followed by adding water to the solid of underlayer and sonicated to form uniform dispersion, repeated above procedures for three times to wash the extra P123 chains. Then the dispersions were centrifuged under 5000 r min$^{-1}$ for 5 min, the upper dispersions were collected this time, which were further centrifuged under 9500 r min$^{-1}$ for 5 min, the underlayer solid was dried under air, ~ 13 mg of nanococrystals were obtained, which gave a yield of ~ 65%. The nanosamples for Per/OFN 1:0, Per/OFN 1:1, Cor/OFN 1:0, and Cor/OFN 1:1 were named as Per NS, Per/OFN NP, Cor NR, and Cor/OFN NR, respectively.

**Calculation of the predicted growth morphologies.** Materials Studio 6.0 software was used to simulate the growth morphologies of Per/OFN and Cor/OFN micrococrystals, based on the attachment energy theory. More specifically, the predicted growth morphology and attachment energies calculations were performed for the Per/OFN and Cor/OFN cocrystals using the universal force field method with ultra-fine quality in the Morphology module of the Material Studio software.

**Imaging and MTT procedures.** Cell imaging studies on cultured MCF-7 cells were conducted on a TSC SPS-II confocal fluorescence microscope (Leica, Germany). Prior to the imaging experiments, the cells were isolated from the culture media and were washed three times with fresh media. Subsequently, the as-obtained cells were incubated for 4 h in fresh media with materials 1–4 (30 µg mL$^{-1}$) at 37 °C. Then, after washed three times with fresh media to remove the free materials, these cells were fixed with 4% paraformaldehyde and subjected to fluorescence imaging experiments.

The cytotoxicity of Per/OFN NP and Cor/OFN NR was evaluated by the standard MTT assay. Briefly, in 96-well U-bottom plates at a density of 7000 cells/well, MCF-7 cells were incubated at 37 °C with different concentrations of nanococrystals for 36 h. Subsequently, the culture media were removed. After adding 0.1 mL of MTT solution (0.5 mg mL$^{-1}$ in fresh media), each well was incubated again at 37 °C for additional 4 h. Followed by discarding supernatant, the as-formed precipitation in each well was dissolved by adding 0.11 mL of DMSO. After shaking the plates for 10 min, the absorbance value (at 490 nm) of each well was recorded on a Microtiter plate assay system. Calculated from the equation: $V_R = A/A_0 \times 100\%$ (where $A$ is the absorbance of the experimental group and $A_0$ is the absorbance of the control group), the cell viability rate ($V_R$) was presented as mean ± standard deviation of five separate measurements.

## Data availability

All data supporting the findings in the article as well as the Supplementary Information files are available from the authors on reasonable request. The X-ray crystallographic coordinates for structures reported in this study have been deposited at the Cambridge Crystallographic Data Centre (CCDC), under deposition numbers 1867077. This data can be obtained free of charge from the Cambridge Crystallographic Data Centre via www.ccdc.cam.ac.uk/data_request/cif.

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

## Acknowledgments

We thank Dr. Guankui Long for the discussion of calculation results and Dr. Tian Dan for structural discussion. Q.Z. acknowledges financial support from AcRF Tier 1 (RG 111/17, RG 2/17, RG 114/16, and RG 8/16) and Tier 2 (MOE 2017-T2-1-021 and MOE 2018-T2-1-070), Singapore. Q.Z. and Z.C. acknowledges financial support from A*STAR funding (SERC 1528000048), Singapore. G.L. thanks the European Union's Horizon 2020 research and innovation program under the Marie Sklodowska-Curie grant agreement No. 791207. H.-L.Z. appreciates the financial support from the National Key R&D Program of China (2017YFA0204903), National Natural Science Foundation of China (NSFC 51733004, 51525303, 221702085, 21673106, 21602093, 21572086, and 1522203), 111 Project and the Fundamental Research Funds for the Central Universities (lzujbky-2017-11 and lzujbky-2017-109). The authors thank beam line BL14B1 (Shanghai Synchrotron Radiation Facility) for providing the beam time.

## Author contributions

Q.Z. led the project and guided the experimental measurements. Y.H. and G.L. conducted the fabrication and characterization of all nano and microparticles. J.X. and Q.G. performed the cell imaging study. L.C. and H.Z. conducted the theoretical calculation. C.Y. perform the Raman study and Z.W. conducted XRD experiment. Q.Z., Z.C. and Y.H. analyzed the data and wrote the paper. All authors discussed the experimental data and provided the valuable comments on the paper.

## Additional information

**Competing interests:** The authors declare no competing interests.

