## [Peer Review File · Nature Communications]

Reviewers' comments:

Reviewer #1 (Remarks to the Author):

This manuscript entitled "Hetero-Aggregation-Induced Enhanced Emission: Reducing Aggregation Caused Quenching (ACQ) Effect through Co-Assembly of PAH Chromophores and Molecular Barriers" presented the micro/nano co-assembly method to solve aggregation caused quenching (ACQ) effect. Octafluoronaphthalene (OFN) was used as molecular barrier, coronene (Cor) and perylene (Per) were chosen as PAHs chromophores, and P123 (PEO20-PPO70-PEO20) was selected as a biocompatible surfactant to protect the micro/nano structures from further growth in aqueous conditions in this paper. Different uniform assemblies (micro/nano wires, sheets, rods and particles) were characterized and studied. The data and analysis of this work are comprehensive. This manuscript is recommended for publication after careful revisions.

1. The authors are talking about the ACQ effect in the title and in the text. However, octafluoronaphthalene (OFN), coronene (Cor) and perylene (Per) were not specified as ACQ molecules in this paper. Please specify.
2. In Page 2, line 30, the authors thought that the corresponding dispersions can be stable at least for 6 months. How is this time period confirmed? Please provide explanation.
3. In Page 4, line 75, the authors presented that OFN was the non-fluorescent molecule. But, in Page 12, Figure 3b, there exists the PL peak of OFN, and the PL intensity is not normalized. Please provide explanation.
4. There should be spaces between numbers and units. In Table 1, there are no spaces between some numbers and units. There also exists this problem in the text. Please revise.
5. In Page 12, line 252, the sentence "that of OFN cocrystals appeared at 351 nm" is wrong. There should be OFN crystal. Please revise. In line 257, the sentence "All the phenomena of blue-shift described...via n-n interaction" is unclear. Please elaborate.
6. In line 261, the PL lifetime of micro Per/OFN cocrystals is 11,450 ns. This value should be wrong. Please revise.
7. In line 295, the sentence "non-fluorescent OFN with a higher band gap of ~ 3.78 eV" is wrong. There should be "band gap". Please revise.
8. In line 299, the sentence "eventually leading to the absence of triplet states" is unclear. Please provide explanation.
9. In cell imaging, Per/OFN NP and Cor/OFN NR could enter the cells mainly because of the role of P123. This has little to do with these three molecules (OFN, Per and Cor). There is nothing special about this cell imaging. Please provide explanation. In Figure 5, line 346, there exists a minor error. Cor NR (j) should be (i).
10. In this work different characterization analysis focuses on different assemblies. There is no focus in the whole text. Please revise.

Reviewer #2 (Remarks to the Author):

This manuscript (ms) submitted for publication to the journal Nature Communications presents an original research work that seems to arise from a recent communication published in the Angew. Chem. Int. Ed. 2018, 57, 1928-1932 (DOI: 10.1002/anie.201712104), submitted among others by two of the authors of this ms and whose subject is totally related to this work.

In this ms It is exposed that two crystalline solid aggregates Per/OFN and Cor/OFN are synthesized from two polycyclic aromatic hydrocarbon (PAH) chromophores: Perylene (Per) and Coronene (Cor), with the non-fluorescent octafluoronaphthalene (OFN) used as molecular barrier in a ratio of 1:1. The synthesis of these crystalline solid aggregates is carried out in the presence of a surfactant P123 at different concentrations to control both the morphology as the scale range of the crystals formed. These two micro/nano cocrystals of Per/OFN and Cor/OFN were characterized by different techniques and was performed an exhaustive study of their improved optical and fluorescence properties.

In order to confirm the detailed molecular stacking in the resultant Per/OFN and Cor/OFN

cocrystals their crystalline structures were characterized in particular by single-crystal x-ray diffraction. The crystal structures presented in this ms are completely correct and agree with many other similar crystalline aggregates deposited in the CCDC database. However, there are some aspects that deserve revision:

- The crystallographic data of Cor/OFN and its crystalline structure presented in this ms had already been previously described in the communication to the journal *Angew. Chem. Int. Ed.* 2018 together with the affine aggregate Ant/OFN in which the anthracene (Ant) was used as PAH chromophore. In that previous publication the crystals of both Ant/OFN and Cor/OFN aggregates were refined using the updated program Shelxl-2014/7, but for the only new crystalline structure of Per/OFN presented in this ms was used the already obsolete refinement program Shelxl-97 which does not include within the resulting crystallographic .cif file neither the instructions used in the refinement nor the structure factors. The deposit of this structure in the CCDC database is not reported in the ms. It is suggested to refine these data with a more up-to-date Shelxl program. The crystallographic data of Per/OFN should be deposited at the CCDC.
- The structural determinations obtained for both Per/OFN and Cor/OFN crystals discussed in the present ms, show at the end of the refinement values of R1 and wR2 higher than usual, which may indicate an insufficient model and/or poor data quality. Also, the Goof (goodness of fit) values of the crystalline structures obtained for both crystals have a higher value than the expected theoretical value close to the unity. In the ms it is indicated that the crystals used for diffraction were not of good quality. Furthermore, by simple inspection of the thermal ellipsoids it is observed that in both structures there is a positional disorder that has not been modelled and whose inclusion in the structural model would allow to lower the values of R1 and wR2 obtained in the refinement. It is suggested to try to include the disorder in the refinement to improve the values of structural agreement.
- On the other hand, for the Cor/OFN crystal data refined in the space group Cm, the CheckCif suggests a greater symmetry than the one used (probably in the space group C2/m). So it would be necessary to examine the refinement in this new space group of greater symmetry to check whether this is the real SG or it is just a pseudosymmetry imposed by the possible positional disorder observed.

In summary, it is my opinion that this ms does not exhibit exceptional novelties to be highlighted with respect to those exposed in the previous communication to the *Angewandte*. In this ms, a customized synthesis of two PAH/OFN aggregates but structurally non-novel and a detailed study of their particular and outstanding optical and fluorescent properties are presented. So I do not recommend its publication in *Nature Communications*, a multidisciplinary journal, but it could be in a journal more specific for its particular subject.

Reviewer #3 (Remarks to the Author):

After careful reading this manuscript, I feel it is a very interesting and meaningful work. As it is known, the planar aromatic structures of π -conjugated metal-free luminescent organic materials generally suffer from aggregation caused quenching (ACQ) effect when they are used in solid state devices or aggregated state, which has greatly impeded their practical applications in optoelectronic devices and biological applications. In this work, the authors cleverly introduced molecule barriers into planar luminescent molecule matrixes and demonstrated a facile, low cost and efficient co-assembly method that avoids complicated molecular projects and obtained uniform micro (can give gram-scale products) or nano cocrystals/co-assemblies (wires, sheets, rods or particles), whose solid PLQYs were greatly enhanced. Protection of PEO chains endows the cocrystals with superb dispersibility in water with high concentrations. More importantly, profiting from above surprising properties, the nano cocrystals presented good cellular permeability, biocompatibility and excellent cell imaging performance. This research is very significant and will provide constructive guidelines to enhance the solid emission of traditional PAH molecules, which normally have ACQ effect in solid state. This success would greatly expand the family of solid emissive materials and open a new avenue for the applications of non- or poor fluorescent PAHs in

bioimaging. The paper is well-written and well-organized.

With my congratulations to the authors' amazing achievement, I highly recommend publishing this paper in Nature Communication after minor revisions.

1. Theoretically, such regularly stacked crystal structures might show diffraction patterns in their SAED results. However, in this work, the authors didn't obtain this data. Any reason?

2. After co-assembling with OFN, the PLQYs of the as-resulted Cor/OFN MW and Cor/OFN NR were enhanced by 235% and 582%, respectively, while the enhancements for the PLQYs of anthracene/OFN and pyrene/OFN were only increased by 97% and 18%, respectively. Comparing with that of Cor/OFN MW and Cor/OFN NR, the increment for the PLQYs of anthracene/OFN and pyrene/OFN are not that high. Any reason?

3. Please indicate if the different standing time would result in different dimension of the co-crystals.

4. To prepare the nano-scale co-crystals, why did the authors use centrifugation procedure instead of suction method, which has been widely used in preparing micro-scale co-crystals?

5. Please indicate if the organic particles have some toxic to bio-systems

6. Some formats of the references did not follow the requirement of this journal, please correct them.

Point-to-point response to reviewers' remarks:

Reviewer #1:

Comment: This manuscript entitled “Hetero-Aggregation-Induced Enhanced Emission: Reducing Aggregation Caused Quenching (ACQ) Effect through Co-Assembly of PAH Chromophores and Molecular Barriers” presented the micro/nano co-assembly method to solve aggregation caused quenching (ACQ) effect. Octafluoronaphthalene (OFN) was used as molecular barrier, coronene (Cor) and perylene (Per) were chosen as PAHs chromophores, and P123 (PEO20-PPO70-PEO20) was selected as a biocompatible surfactant to protect the micro/nano structures from further growth in aqueous conditions in this paper. Different uniform assemblies (micro/nano wires, sheets, rods and particles) were characterized and studied. The data and analysis of this work are comprehensive. This manuscript is recommended for publication after careful revisions.

Reply: We thank the reviewer for carefully reading our work and providing constructive feedback.

Comment (1). The authors are talking about the ACQ effect in the title and in the text. However, octafluoronaphthalene (OFN), coronene (Cor) and perylene (Per) were not specified as ACQ molecules in this paper. Please specify.

Reply: Thanks for the referee's comment. OFN, Cor and Per with ACQ effect have been specified in page 3.

Comment (2). In Page 2, line 30, the authors thought that the corresponding dispersions can be stable at least for 6 months. How is this time period confirmed? Please provide explanation.

Reply: Thanks for the reviewer's comments. We initially confirmed this period via the observation of the dispersions by naked eye and no visual precipitation was found after 6 months (Figure R1). The stability of the dispersions was further confirmed by the nearly unchanged morphologies/dimensions (Figure R2), PL spectra, and PLQYs (Figure R3), by comparing with that of the freshly prepared ones. Figure R1, R2 and R3 have been added in SI (page 22 and 24).

Figure R1. Dispersion photos of Per/OFN NP (a) and Cor/OFN NR (b), respectively, after being aged for 6 months (Concentration of the dispersions, $1 \text{ mg}\cdot\text{mL}^{-1}$). After being aged for 6 months, the two dispersions were still uniform, and no visualized precipitation appeared.

Figure R2. AFM images of Per/OFN NP (a) and Cor/OFN NR (b), respectively, after being aged for 6 months. After being aged for 6 months, the dimensions of Per/OFN NP and Cor/OFN NR were still smaller than 200 nm, which confirmed the stability of the morphologies of the nano cocrystals.

Figure R3. PL emission spectra of the water dispersions for Per/OFN NP (a) and Cor/OFN NR (b), respectively, after 6 months placement. As well as the photographs of the water dispersions (concentration of the dispersions, $1 \text{ mg}\cdot\text{mL}^{-1}$) under irradiation of 365 nm laser after 6 months (insets). The PL spectra of the two nano cocrystals presented the same location and shape as that of the freshly prepared ones demonstrated in Figure S26(b) and S27(b). The PLQYs of aged Per/OFN NP and Cor/OFN NR were 24.8% and 22.6%, which were close to that of the freshly prepared ones (26.4% and 23.2%, respectively), further confirmed the stability of the dispersions.

Comment (3). In Page 4, line 75, the authors presented that OFN was the non-fluorescent molecule. But, in Page 12, Figure 3b, there exists the PL peak of OFN, and the PL intensity is not normalized. Please provide explanation.

Reply: Thank for the referee's valuable comments. It is our mistake to give an incorrect definition for OFN as non-fluorescent molecule. In the revised manuscript, we have corrected it to "OFN with weak fluorescence" because its solid PLQY was tested to be 2.6%, which was specified in page 3, 4 and page 13. To clearly compare the peak locations of the three curves for OFN crystal, PAH crystal and cocrystal, respectively, we employed their highest intensities as 1 to decrease the interference from different peak heights, which was indicated in Figure 3 caption.

Comment (4). There should be spaces between numbers and units. In Table 1, there are no spaces between some numbers and units. There also exists this problem in the text. Please revise.

Reply: Thank for the referee's comments. In the revised manuscript, spaces were added between every numbers and units except for number and % as well as number and °C.

Comment (5). In Page 12, line 252, the sentence "that of OFN cocrystals appeared at 351 nm" is wrong. There should be OFN crystal. Please revise. In line 257, the sentence "All the phenomena of blue-shift described...via π - π interaction" is unclear. Please elaborate.

Reply: Thank for the referee's valuable comments. "that of OFN cocrystals appeared at 351 nm" has been corrected into "that of OFN crystals appeared at 351 nm" (page 11). In addition, we thank the reviewer's comments that "All the phenomena of blue-shift described...via π - π interaction" is unclear. The periodic OFN molecular barriers could cause the blue-shifted absorptions of cocrystals with respect to pristine crystals. Also, time-dependent density functional theory calculations presented in our former research (Angew. Chem. Int. Ed. 2018, 57, 1928 –1932) can further prove this point. The PL blue-shift is a consequence of the screening of the π -interactions between PAH-PAH by OFN, which can decrease the exciton-delocalization-induced PL red-shifts (J. Chem. Phys. 2008, 128, 054505). To make it clear in our manuscript, the detailed explanation has been added in page 11-12.

Comment (6). In line 261, the PL lifetime of micro Per/OFN cocrystals is 11,450 ns. This value should be wrong. Please revise.

Reply: We appreciate the reviewer's comments. We recalculate the PL lifetime of Per/OFN cocrystal via the common equation (R1) according to the original data showed below for several times. The lifetime (11,450 ns) of Per/OFN cocrystal is correct, about ~ 11.5 μ s. We also have changed some difference in revised manuscript.

T1	=	62.96752	ch;	6.910016E-09	sec	S.Dev = 9.700925E-11	sec
T2	=	132.4109	ch;	1.453069E-08	sec	S.Dev = 1.361146E-10	sec
T3	=	106191.2	ch;	1.165335E-05	sec	S.Dev = 2.933462E-06	sec
A	=	-199.3864				S.Dev = 14.57143	
B1	=	3780.093	[1.07 Rel.Ampl][0.74 Alpha]			S.Dev = 16.1485	
B2	=	1125.909	[0.67 Rel.Ampl][0.22 Alpha]			S.Dev = 7.755672	
B3	=	205.0439	[98.25 Rel.Ampl][0.04 Alpha]			S.Dev = 14.77122	

$$\tau = \frac{\sum B_i T_i^2}{\sum B_i T_i} \quad (R1)$$

where B_i is the fitting constant, T_i is the lifetime of different decay models. The fitting results were shown in Table S5.

Comment (7). In line 295, the sentence “non-fluorescent OFN with a higher bad gap of ~ 3.78 eV” is wrong. There should be “band gap”. Please revise.

Reply: Thank for the reviewer's comment. “non-fluorescent OFN with a higher bad gap of ~ 3.78 eV” has been corrected as “weakly fluorescent OFN with a higher band gap of ~ 3.78 eV”.

Comment (8). In line 299, the sentence “eventually leading to the absence of triplet states” is unclear. Please provide explanation.

Reply: Thank for the reviewer's very valuable suggestion. High-energy-gap OFN molecular barriers were intercalated in a periodic π -stacked molecular array, which effectively cut off the interaction of the low-energy-gap PAH chromophores (as presented in Figure S11, the distances between two PAH planes in Per/OFN cocrystals was increased to 6.886 Å by the intercalation of OFN molecules. Similarly, as shown in Figure S13, the distances between two PAH planes in Cor/OFN cocrystals were increased to 6.872 Å.), eventually leading to the absence of triplet states (Angew. Chem. Int. Ed. 2018, 57, 1928 –1932). We can understand this from the below schematic, which has been added to the revised supporting information.

Figure R4. Schematic of interactions between OFN and PAHs.

Comment (9). In cell imaging, Per/OFN NP and Cor/OFN NR could enter the cells mainly because of the role of P123. This has little to do with these three molecules (OFN, Per and Cor). There is nothing special about this cell imaging. Please provide explanation. In Figure 5, line 346, there exists a minor error. Cor NR (j) should be (i).

Reply: Thank for the reviewer’s valuable comments. Indeed, Per/OFN NP and Cor/OFN NR could enter the cells mainly because of the role of P123, which has good biocompatibility. We believe that the usage of P123 could endow PAH molecules (which have no cellular compatibility or even high cytotoxicity) with much better biological compatibility and lay the foundation for further biological research of such PAH-based materials. For the cell imaging experiments, it was a preliminary exploration that such kind of nano fluorescent materials based on PAH could be distributed evenly in cells instead of aggregation, which also pave the way for further intracellular experiments. Moreover, we made a further confirmation via cell imaging. Comparing to single-component crystals, OFN-doped cocrystals still presented an enhanced fluorescence even in the complex environment of cells. Therefore, such kind of fluorescent materials is expected to be used as alternate materials for future disease diagnosis and therapy via cell imaging.

In Figure 5, “Cor NR (j)” has been corrected to “Cor NR (i)”.

Comment (10). In this work different characterization analysis focuses on different assemblies. There is no focus in the whole text. Please revise.

Reply: Thank for the reviewer’s valuable suggestions. We agree with this point and have done our best to shorten the morphology characterizations part. This work mainly focuses on preparation, characterization and application of fluorescent materials with “Hetero-Aggregation-Induced Emission Enhancement: Reducing Aggregation Caused Quenching (ACQ) Effect”. The materials were prepared through the co-assembly of PAH chromophores

and OFN molecular barriers, while for a self-assembled material, detailed characterization for their morphologies as well as the way the molecules stacked are essential. In addition, the method we used here not only is a facile, low cost and efficient micro/nano co-assembly method but also can well control the dimensions of resulted cocrystals, which could be better to meet the requirements of various applications of such kind of aggregated luminescent materials and can be considered to lay a foundation for their applications. Thus, the characterizations of the assemblies are not only one of the important characterizations for co-assembled luminescent materials but also closely related to further applications, which are very important. However, this part might be too long, so we have shortened the morphology characterizations part to 1.5 pages.

Reviewer #2 (Remarks to the Author):

Comment: This manuscript (ms) submitted for publication to the journal Nature Communications presents an original research work that seems to arise from a recent communication published in the Angew. Chem. Int. Ed. 2018, 57, 1928-1932 (DOI: 10.1002/anie.201712104), submitted among others by two of the authors of this ms and whose subject is totally related to this work. In this ms, It is exposed that two crystalline solid aggregates Per/OFN and Cor/OFN are synthesized from two polycyclic aromatic hydrocarbon (PAH) chromophores: Perylene (Per) and Coronene (Cor), with the non-fluorescent octafluoronaphthalene (OFN) used as molecular barrier in a ratio of 1:1. The synthesis of these crystalline solid aggregates is carried out in the presence of a surfactant P123 at different concentrations to control both the morphology as the scale range of the crystals formed. These two micro/nano cocrystals of Per/OFN and Cor/OFN were characterized by different techniques and was performed an exhaustive study of their improved optical and fluorescence properties. In order to confirm the detailed molecular stacking in the resultant Per/OFN and Cor/OFN cocrystals their crystalline structures were characterized in particular by single-crystal x-ray diffraction. The crystal structures presented in this ms are completely correct and agree with many other similar crystalline aggregates deposited in the CCDC database. However, there are some aspects that deserve revision:

Reply: We thank the reviewer for carefully reading our work and providing constructive feedback.

Comment (1): The crystallographic data of Cor/OFN and its crystalline structure presented in this ms had already been previously described in the communication to the journal *Angew. Chem. Int. Ed.* 2018 together with the affine aggregate Ant/OFN in which the anthracene (Ant) was used as PAH chromophore. In that previous publication the crystals of both Ant/OFN and Cor/OFN aggregates were refined using the updated program Shelxl-2014/7, but for the only new crystalline structure of Per/OFN presented in this ms was used the already obsolete refinement program Shelxl-97 which does not include within the resulting crystallographic cif file neither the instructions used in the refinement nor the structure factors. The deposit of this structure in the CCDC database is not reported in the ms. It is suggested to refine these data with a more up-to-date Shelxl program. The crystallographic data of Per/OFN should be deposited at the CCDC.

Reply: Thank for the reviewer's valuable comments. We redid the refinement of the structure of Per/OFN including the disorder using the up-to-date Shelxl program- Shelxl-2014. The Cor/OFN and Per/OFN structures have already deposited at the CCDC with numbers of 1575415 and 1867077, respectively.

Comment (2): The structural determinations obtained for both Per/OFN and Cor/OFN crystals discussed in the present ms, show at the end of the refinement values of R1 and wR2 higher than usual, which may indicate an insufficient model and/or poor data quality. Also, the Goof (goodness of fit) values of the crystalline structures obtained for both crystals have a higher value than the expected theoretical value close to the unity. In the ms it is indicated that the crystals used for diffraction were not of good quality. Furthermore, by simple inspection of the thermal ellipsoids it is observed that in both structures there is a positional disorder that has not been modelled and whose inclusion in the structural model would allow to lower the values of R1 and wR2 obtained in the refinement. It is suggested to try to include the disorder in the refinement to improve the values of structural agreement.

Reply: Thank for the reviewer's valuable comments. We totally agree with the reviewer's opinion about the problem of the crystal structures. Due to poor crystal quality (we have tried several crystals), twinning and disorder, we could not obtain good diffraction data and especially at the high resolution. We redid the refinement of the structure of Per/OFN including the disorder using the up-to-date Shelxl program- Shelxl-2014. The results indicated that there are come decrease of R1, Goof and wR2 values as follows: The Goof value changes from 1.297 to 1.123, the R1 value changes from 0.1120 to 0.1087 and wR2 values change

from 0.3204 to 0.3178 respectively. The related data have updated at the new Table S2 in the SI.

Comment (3): On the other hand, for the Cor/OFN crystal data refined in the space group Cm, the CheckCif suggests a greater symmetry than the one used (probably in the space group C2/m). So it would be necessary to examine the refinement in this new space group of greater symmetry to check whether this is the real SG or it is just a pseudosymmetry imposed by the possible positional disorder observed.

Reply: We appreciate the reviewer's valuable suggestions. For checking the question of pseudosymmetry of Cor/OFN. A powder SHG (second harmonic generation) measurement was carried out by Cor/OFN crystalline powder at room temperature. The obvious SHG signals demonstrate its nonlinear optical (NLO) behavior (Figure R3), which indicates that the Cor/OFN has a noncentrosymmetrical feature at room temperature, i.e. the space group Cm is correct space group and the C2/m is just a pseudosymmetry. Experimental section: The SHG measurement was performed using a Cr, Tm, Ho: YAG laser beam ($\lambda = 2.09 \mu\text{m}$) by Cor/OFN crystalline powder at room temperature.

Figure R5. The comparative SHG oscilloscope traces of the Cor/OFN crystalline powder.

Other comment for Reviewer #2: In summary, it is my opinion that this ms does not exhibit exceptional novelties to be highlighted with respect to those exposed in the previous communication to the Angewandte. In this ms, a customized synthesis of two PAH/OFN aggregates but structurally non-novel and a detailed study of their particular and outstanding optical and fluorescent properties are presented. So I do not recommend its publication in Nature Communications, a multidisciplinary journal, but it could be in a journal more specific for its particular subject.

Reply: Thank very much for the reviewer's valuable comments. Although one (Cor/OFN) of the co-crystal structures here has been published in our previous work for *Angewandte*, this research is totally different from the previous one for the following reasons:

According to our humble opinion, in the previous research, although we have already found that the intercalation OFN molecular barrier can enhance the PLQY of coronene and anthracene and used various physical characterizations to elucidate the underlying mechanisms. It is just a preliminary study, no other expanding or deep studies (especially the fluorescent properties and further applications which are essential for fluorescent materials) hasn't been made due to limited space of the manuscript for *Angewandte*. For example, this method is very good, but we still wonder if it works when OFN is doped into the system of other polycyclic aromatic hydrocarbon (PAH) chromophores (such as pyrene and perylene) that have low PLQYs in solid. Moreover, it is very important to confirm if the PLQY still can be enhanced after reducing the dimensions of the fluorescent cocrystals to micro/nano scales, because nano/micro materials have huge potential applications in optoelectronic fields and bio-imaging. Especially, as we all know, we never thought that hydrophobic PAHs with high toxicity can be used in bioimaging in the past. If all above properties were confirmed, it will greatly expand the scope of applications of such method and the as-resulted fluorescent materials. That's key points that we are doing here.

Although we used a customized synthesis way to prepare two PAH/OFN aggregates, the method was facile, low cost and efficient, which can avoid complicated molecular projects and easily control the dimensions of the resulted cocrystals. To verify the universality of this method, perylene and pyrene that haven't been studied in previous *Angewandte* were also investigated in this work. The results indicated that the fluorescence of perylene and pyrene were also improved, which further demonstrated the universality of the method presented in this work. More importantly, the protection of P123 endowed these cocrystals with superb dispersibility in water with high concentration ($10 \text{ mg} \cdot \text{mL}^{-1}$), which made the nano cocrystals display excellent biocompatibility, cellular permeability, low toxicity, and wonderful bioimaging. Therefore, this research has a great significance and will provide constructive guidelines for fabricating highly solid emissive materials using traditional ACQ molecules (PAHs) as raw materials. In addition, our success will greatly expand the family of solid emissive materials, and open novel avenues for the applications of originally non- or poor fluorescent PAHs (ACQ effect in aggregated state) in bioimaging or optical devices.

Reviewer #3 (Remarks to the Author):

Comment: After careful reading this manuscript, I feel it is a very interesting and meaningful work. As it is known, the planar aromatic structures of π -conjugated metal-free luminescent organic materials generally suffer from aggregation caused quenching (ACQ) effect when they are used in solid state devices or aggregated states, which has greatly impeded their practical applications in optoelectronic devices and biological applications. In this work, the authors cleverly introduced molecule barriers into planar luminescent molecule matrixes and demonstrated a facile, low cost and efficient co-assembly method that avoids complicated molecular projects and obtained uniform micro (can give gram-scale products) or nano cocrystals/co-assemblies (wires, sheets, rods or particles), whose solid PLQYs were greatly enhanced. Protection of PEO chains endows the cocrystals with superb dispersibility in water with high concentrations. More importantly, profiting from above surprising properties, the nano cocrystals presented good cellular permeability, biocompatibility and excellent cell imaging performance. This research is very significant and will provide constructive guidelines to enhance the solid emission of traditional PAH molecules, which normally have ACQ effect in solid state. This success would greatly expand the family of solid emissive materials and open a new avenue for the applications of non- or poor fluorescent PAHs in bioimaging. The paper is well-written and well-organized. With my congratulations to the authors' amazing achievement, I highly recommend publishing this paper in Nature Communication after minor revisions.

Reply: We thank the reviewer for carefully reading our work and providing constructive feedback.

Comment (1). Theoretically, such regularly stacked crystal structures might show diffraction patterns in their SAED results. However, in this work, the authors didn't obtain this data. Any reason?

Reply: Thank for the valuable reviewer's comments. We attempted to elucidate the crystal structure of the cocrystals by electron diffraction (ED), however, no any crystal lattice (Figure S16c and 18c) and diffraction patterns (Figure S16d and 18d) were observed, which might be attributed to the highly energetic irradiation of electron beams that can destroy crystal structures (Egerton, R. F.; Malac, P.; Li, M. *Micron* 2004, 35, 399). The explanation has been added in the revised manuscript.

Comment (2). After co-assembling with OFN, the PLQYs of the as-resulted Cor/OFN MW and Cor/OFN NR were enhanced by 235% and 582%, respectively, while the enhancements for the PLQYs of anthracene/OFN and pyrene/OFN were only increased by 97% and 18%, respectively. Comparing with that of Cor/OFN MW and Cor/OFN NR, the increment for the PLQYs of anthracene/OFN and pyrene/OFN are not that high. Any reason?

Reply: Thank for the valuable reviewer's comments. We have tried most of the PAH molecules to increase their PLQY via the co-assembly with OFN molecular barrier. We found that the enhancement of PLQY is more efficient for the PAHs with much higher PLQY while the materials with weaker ACQ effect in solid have small enhancement. This might be due to their inherent better fluorescent properties. As we all know, it is easier to make an improvement based on a poor one, but much difficult to improve an inherently good one.

Comment (3). Please indicate if the different standing time would result in different dimension of the co-crystals.

Reply: We appreciate the reviewer's valuable suggestions. The dimensions of the assemblies can be stable for 6 months (Figure R2). In addition, we checked the stability of the nano-scale suspensions via observation by naked eye, AFM visualization, PL spectra and PLQYs. After 6 months, no visual precipitation formed (Figure R1), and the AFM results of the dispersions still presented well dispersed nano particles almost with the same dimensions as that of the freshly prepared nano crystals (Figure R2). The PL spectra (Figure R3) of the two nano cocrystals presented the same location and shape as that of the freshly prepared ones demonstrated in Figure S26(b) and S27(b). The PLQYs of aged Per/OFN NP and Cor/OFN NR were 24.8% and 22.6%, which were close to that of the freshly prepared ones (26.4% and 23.2%, respectively), further confirmed the stability of the dispersions.

Comment (4). To prepare the nano-scale co-crystals, why did the authors use centrifugation procedure instead of suction method, which has been widely used in preparing micro-scale co-crystals?

Reply: Thank for the valuable reviewer's comments. On one hand, in the beginning, we have tried to collect the nano-scale crystals, however, the dimensions of the crystals were too small to be filtered quickly. Even some of the nano particles could come into the filtrate, which would lead to a low yield. On the other hand, we can purify the nano co-crystals via gradient centrifugation procedures to make sure that the dimensions of the assemblies were uniform and suitable for cell penetration.

Comment (5). Please indicate if the organic particles have some toxic to bio-systems.

Reply: We appreciate the reviewer's valuable suggestions. In this work, the *vitro* cytotoxicity of the four nano crystals was evaluated *via* the standard MTT (3-(4,5-dimethylthiazol-2-yl)-2,5-diphenyltetrazolium bromide) assay using MCF-7 cells (Figure S27). As anticipated, low cytotoxicity of such nano crystals was confirmed. Specifically, over 90% cell viability was observed after incubating MCF-7 cells with any of the four crystals with different concentrations ranging from 0 to 50 $\mu\text{g}\cdot\text{mL}^{-1}$, which was indicated in page 15.

Comment (6). Some formats of the references did not follow the requirement of this journal, please correct them.

Reply: Thank for the reviewer's comment. All the references have been checked and corrected.

Other revisions:

- (1) In the acknowledgement part, "Z. C. acknowledges financial support from A*STAR funding (SERC 1528000048), Singapore" and "HLZ appreciate the financial support from the National Key R&D Program of China (2017YFA0204903), National Natural Science Foundation of China (NSFC. 51733004, 51525303, 221702085, 21673106, 21602093, 21572086, 1522203), 111 Project and the Fundamental Research Funds for the Central Universities (lzujbky-2017-11, lzujbky-2017-109). The authors thank beam line BL14B1 (Shanghai Synchrotron Radiation Facility) for providing the beam time." were added.
- (2) At the end of the manuscript, entry for the Table of Contents was added.

REVIEWERS' COMMENTS:

Reviewer #1 (Remarks to the Author):

I have gone through the revised manuscript and found that the authors have adequately addressed my previous comments and suggestions. The revisions are satisfactory and the changes are acceptable. The quality of the manuscript has been improved after revision. I do not have further criticism of the work.

Reviewer #2 (Remarks to the Author):

I have carefully read the point-by-point response letter, as well as the revised manuscript and I can tell you that the points raised in the previous round of review have been addressed and answered by the authors in an admirably satisfactory manner. So I am willing to change my initial reservation and recommend the publication of the manuscript entitled: "Hetero-Aggregation-Induced Enhanced Emission: Reducing Aggregation Caused Quenching (ACQ) Effect through Co-Assembly of PAH Chromophores and Molecular Barriers" by Prof Zhang and colleagues, in Nature Communication.

Reviewer #3 (Remarks to the Author):

The authors answer the questions already, please publish as it.